# Diversity in Landscape Management Affects Butterfly Distribution

**Katarzyna Szyszko-Podgórska** [1,*] **, Izabela Dymitryszyn** [2] **and Marek Kondras** [3]

1   The Institute of Environmental Protection—National Research Institute, Słowicza 32, 02-170 Warsaw, Poland
2   Department of Landscape Art, Institute of Environmental Engineering, Warsaw University of Life Sciences—WULS, 02-787 Warsaw, Poland; izabela_dymitryszyn@sggw.edu.pl
3   Department of Soil Sciences, Institute of Agriculture, Warsaw University of Life Sciences—WULS, 02-787 Warsaw, Poland; marek_kondras@sggw.edu.pl
*   Correspondence: katarzyna.podgorska@ios.edu.pl

**Abstract:** The aim of the study was to determine the influence of differentiated landscape management on the distribution and abundance of butterfly species. The question was raised with regard to which land use type benefited butterflies, and which affected them, that is: under which management type does biodiversity increase, and under which is it depleted? The spatial and abundance distributions of the examined butterfly species diverged considerably. The observed differences between the abundance distributions may be due to diverse conditions in the small-scale environments or specific food preferences of individual species. The diversified management of the "Krzywda" landscape fosters the abundance of mesophilic and ubiquitous butterfly species, whereas xerotermophilic and hygrophilous species are not fostered. The transects established on the fallow land with harvested biomass as well as that with unharvested biomass and in the forest ecotone showed that the fallows were characterized by the highest abundance of butterflies, and the greatest number of plant species was recorded there. Mown fallow lands with not harvested biomass as well as forested areas fostered polyphagous and monophagous butterfly species. Oligophagous butterfly species were fostered by mown fallow lands with not harvested biomass. Unmown meadows, the ecotone marshland and fallow, as well as unmown fallow lands did not foster butterfly abundance. This most likely means that land management can influence the food base of butterflies, and consequently, their abundance. The stage of succession as well as the homogeneity of the area in terms of vegetation had the strongest filtering effect on the occurrence and distribution of butterflies among the analyzed variables. The number of species as well as their abundance was higher on transects classified as young successional stages on which successional processes were artificially inhibited by mowing and biomass removal. Advanced environmental engineering enables humans to influence species composition in a given ecosystem to achieve a desired result. There is no doubt that human activity will be successful when the needs of individual species in a given environment are accurately understood.

**Keywords:** lepidoptera; landscape; biodiversity; management

## 1. Introduction

One of the most important problems of highly developed countries is the rapid loss of species. The main cause of the loss in the variety of life is believed to be anthropogenic changes within the environment, including the degradation of natural resources in the landscape system. People, through their economic activities related to agriculture, industry, infrastructure and water drainage, have affected the appearance and state of the natural environment. Human activities not only deter many native species from altered environments, but also lead to the permanent isolation of groups of many species, and thus time after time, increasing the risk of their extinction [1–4]. Humans, by shaping spatial patterns, can create proper conditions for the occurrence of various animal species. Animals are

able to move freely, hence, they can be good indicators of microclimatic conditions created for them [5,6]. Consequently, humans have the potential to influence the occurrence of native animal species, inter alia, by managing spatial patterns in such a way that conditions are fostered for the occurrence of expected species. The incidence of animals is, on the one hand, a testimony to our actions in a given territory, and, on the other, informs us whether we have structured the area merely scenically or also functionally. At this point, it is important to note that the occurrence of species in a given area changes over time and is related to ecological succession, which can be defined as an ordered and interacting process of changes taking place in the biocoenosis and abiotic environment [7,8]. The biocoenoses following one another on the same site are in successional stages. We can derive various benefits from each biocoenosis developmental stage, and through hindering or stimulating the developmental processes, we can multiply these benefits. Each species requires a specific habitat as the living environment [9–11]. In the case of animals, the sites are used in a range of ways, depending on diverse locations, which can serve as the areas that provide food or reproduction sites, or else a wildlife refuge. The species specific to particular biocoenosis successional stages use the area within the boundaries of a given stage. The absence of any of such areas inevitably leads to a decrease in population numbers [12]. There is currently an exponential decline in insect biodiversity [13–15], which is probably caused by environmental management [5].

## 2. Objectives and Scope of the Study

The study aimed at evaluating human activities in terms of their effects on butterflies with regard to their species distribution and abundance. The questions were raised as to which of the examined land uses positively influenced the abundance of butterflies and which had adverse impacts; moreover, which landscape management activities increased biodiversity, and which depleted it? What measures need to be taken to ensure the incidence of the butterfly species we want to be present in the landscape shaped by humans, and how to manage the abundance of these species?

Previous studies at the site have shown differences in soil and plant characteristics among a number of selected ecosystems and ecotones, as well as differences in beetle and butterfly communities [5]. They showed that beetles and butterflies showed different responses to different features of the various sites studied, which can be explained by differences in relation to their ecological characteristics, such as feeding preferences. Therefore, an attempt was made to focus more in detail on factors affecting the qualitative and quantitative occurrence of butterflies in given habitats.

We formulated the following hypotheses:

1.  Butterflies show differences in response to land use form.
2.  Butterflies show differences in area selection according to food preferences and habitat preferences.
3.  Quantitative occurrence of melliferous vegetation (number of melliferous plant) affects the occurrence of butterflies.
4.  The stage of succession affects the quantitative occurrence of butterflies and butterfly biodiversity.

In order to achieve the assumed study objectives, the species composition of butterflies and their spatial distribution were determined depending on the ecosystem and ecotone types under different management systems. The occurrence of butterflies was analyzed in relation to food preferences, habitat requirements in land use forms, as well as sex of the butterflies under the study.

The study was conducted in northwestern Poland (Zachodniopomorskie Voivodeship), in the buffer zone of the Drawieński National Park (the Forest District Tuczno). The study area was the research site "Krzywda", comprising various habitats of cultivated fields, fallow land, forests and marshes. The object was established in 1993 and covers more than 172 ha of fallow land, pastures and set-aside lands, as well as approximately 68 ha of marshland subject to heavy eutrophication caused by human economic activity. The

marshland is fed by three watercourses [16]. In the area, research has been carried out with the aim to record successional changes in fauna and flora, which are, among others, due to human economic activities. The launch of the sewage treatment plant in the city of Tuczno and the discharge of treated wastewater into the polluted marshland have allowed us to observe changes in the species composition of this water reservoir. Uncultivated agricultural lands, pastures and meadows, differentiated in terms of soil fertility (poor soils of the quality classes III–VI), were subjected to the study with regard to the non-natural inhibition of the succession processes through cutting off and removing the plant matter out of the site [16]. In our study, seven sites were included, which were three fallow lands, two meadows under different management modes and two ecotones with different ecosystems (Figure 1, Table 1).

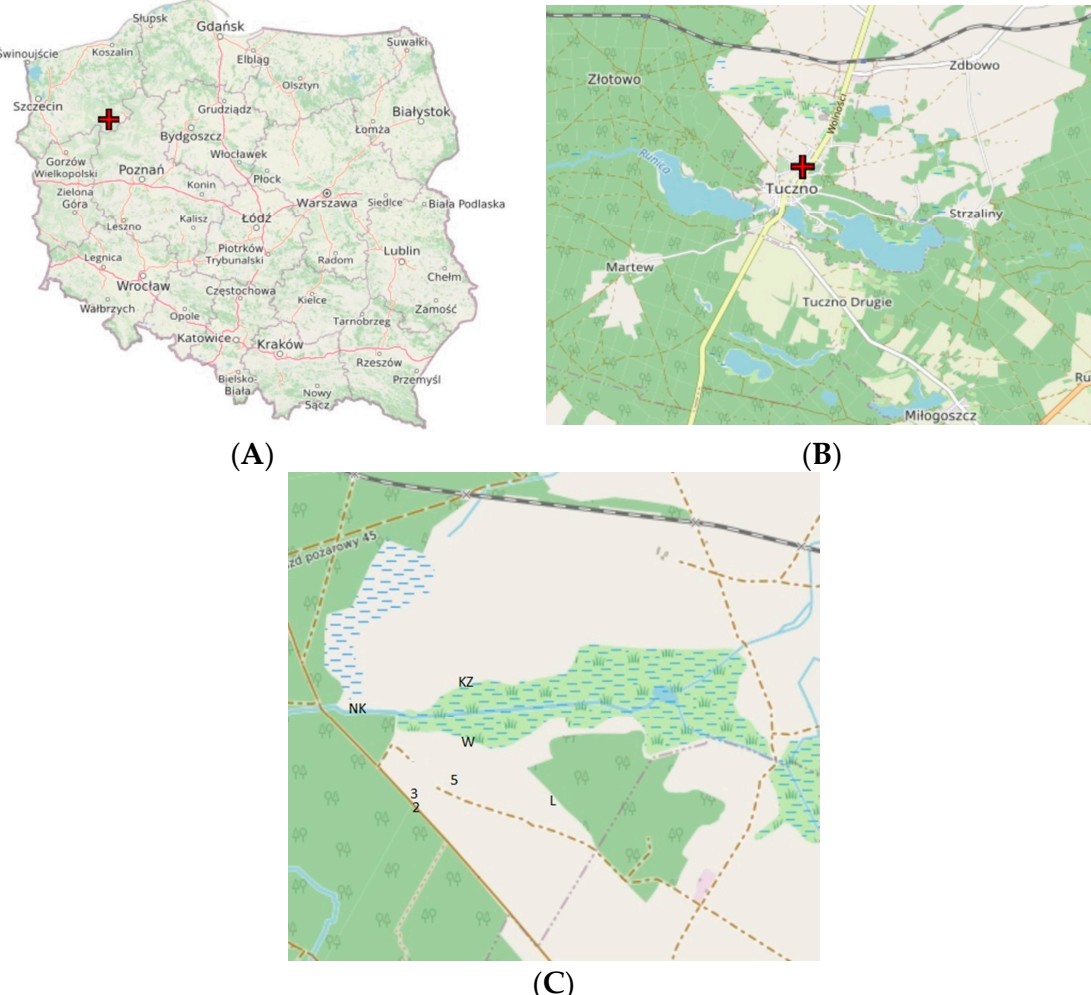

**Figure 1.** Position of research transects in the Tuczno area: (**A**) Location of Tuczno in Poland; (**B**) location of research project; (**C**) location of study transects (labeled as in Table 1).

**Table 1.** Types of the study transects and their descriptions, including information with regard to dominant plant species (% share in total plant cover) and each transect area (ha as of 2018) (Source: [5] revised).

| Transect No. | Type | Description | Area (ha) | Plant Cover (%) | Dominant Plant Species | Bray–Curtis Plant Similarity | Stage of Succession | No. of Plant Species | No. of Melliferous Plant Species |
|---|---|---|---|---|---|---|---|---|---|
| 2 | Fallow land | Mown post-agricultural land with biomass not removed | 3.2 | 1.86 | *Anthoxanthum odoratum, Pleurozium schreberi, Holcus lanatus, Deschampsia flexuosa* | 0.503 ± 0.100 | 2 | 50 | 25 |
| 3 | Fallow land | Mown post-agricultural land with biomass removed | 5.7 | 3.31 | *Anthoxanthum odoratum, Hieracium pilosella, Festuca rubra, Armeria elongata* | 0.599 ± 0.088 | 1 | 44 | 23 |
| 5 | Fallow land | Unmown post-agricultural land | 10.8 | 6.28 | *Anthoxanthum odoratum, Pleurozium schreberi, Deschampsia flexuosa, Phleum pretense* | 0.511 ± 0.094 | 3 | 44 | 26 |
| W | Ecotone | Ecotone marshland -fallow | 2.5 | 1.45 | *Agrostis capillaris, Arrhenatherum elatius, Festuca rubra, Phalaris arundinacea* | 0.366 ± 0.157 | 4 | 39 | 21 |
| L | Ecotone | Ecotone forest -fallow | 0.9 | 0.52 | *Sarothamnus scoparius, Anthoxanthum odoratum, Pinus silvestris, Agrostis capillaris* | 0.550 ± 0.101 | 4 | 66 | 40 |
| NK | Meadow | Unmown meadow | 6.8 | 3.95 | *Festuca rubra, Pleurozium schreberi, Arrhenatherum elatius, Agrostis capillaris* | 0.400 ± 0.188 | 3 | 42 | 28 |
| KZ | Meadow | Mown meadow | 2.6 | 1.51 | *Agrostis capillaris, Arrhenatherum elatius, Anthoxanthum odoratum, Dactylis glomerata* | 0.474 ± 0.099 | 2 | 54 | 25 |

## 3. Materials and Methods

### 3.1. Field Methods

In each of the seven studied sites, a study transect of approximately 800 m length were established. Six sample plots were established on each transect. Within each of these transects, butterflies were caught with the use of an insect net. The recorded butterflies were assigned to the nearest sampling plot. The line transects were delineated consistently with the methodology adopted in butterfly surveys [17,18]. The linear transect was defined as a 5 m wide strip of land, along which the observer moved at a slow pace (about 3 km/h) to catch and record adult butterflies. The observations were carried out in the years 2020–2022, from May to September, twice a month, during one day. The observation months were selected based on the assumption that the best weather conditions for butterfly emergence were in place at the time. The terminology used was consistent with Buszko and Masłowski [19].

Six 5 m × 5 m sample plots for phytosociological surveys [20] were designated on each study transect. A total of 42 surveys were carried out on 14–16 June 2020. The surveys were elaborated by recording the species and describing their occurrence with the use of the

Braun-Blanquet Plant Species Cover-Abundance Scale [20]. Plant species were determined based on the nomenclature by Mirek et al. [21]. The total cover of each vegetation layer in a given sample transect was assessed under field conditions. For each phytosociological survey, plant species cover values for particular recorded plant species were converted to the average percentage cover value using the Braun-Blanquet scale: "+"—0.1%, 1—5%, 2—17.5%, 3—37.5%, 4—62.5% and 5—87.5%. The mean percentage covers of species for a given sample transect (survey) were summed to determine the species dominance value (%) for each survey transect [5].

*3.2. Data Analysis*

Based on the material collected in 2020–2022, where the data from six sample plots of three years were pooled for each transect, the dominance structure of butterfly assemblages was determined. The classification adopted a division of the dominance distribution into 6 classes: eurecedent (I, eR), recedent (II, R), subrecedent (III, sR), subdominant (IV, sD), dominant (V, D) and eudominant (VI, eD) [22]. The dominant structure was calculated using the formula:

$$a = \sqrt[6]{n}$$

where a—the limit value for the lowest class, obtained by the sixth degree root of a number equal to the abundance of the most numerous species. The limits of next classes were determined by the values of successive powers of the value 'a'.

The proposed division into dominance classes depended on the sum of individuals occurring in a given study transect, and was concluded by comparing the abundance of a given butterfly species with that of the most abundant species [22]. The division allowed for the evaluation of the dominance structure. In the classification assumed, the term "dominant" referred to the dominant species and the term "recedent" to the receding species. This approach made it possible to appraise the role of individual species in the butterfly assemblages examined, as well as requirements of individual species in relation to the form of land use.

Then, the frequency of the species was calculated, which informs us about the commonness or rarity of the species. Frequency was calculated according to the formula [23]:

$$Fi = (s/S) \times 100\%$$

where:

Fi—Frequency of a given species;
s—Number of transects with a given species;
S—Number of all transects.

The Shannon–Weiner Species Diversity Index *H* was calculated according to the formula by Shannon and Wiener [24]:

$$H = \sum_{i=1}^{S} p_i log p_i$$
$$p_i = \frac{n_i}{N_i}$$

where $n_i$ is the number of specimens of a specific species and $N_i$ is the number of all specimens of all the species.

The collected research material was assigned to the ecological groups. This was performed in order to assign butterfly species to the groups of organisms that have a similar range of tolerance to environmental factors, and to help determine whether these preferences had a significant influence on the distribution of butterflies in the "Krzywda" area. Therefore, based on the literature data, the butterfly species under the study were characterized in terms of their trophic index (It) and ecological index (Ie) and assigned into relevant groups [19,25–57]. The trophic index was adjusted using three species-specific trophic levels: 1 = monophagy; 2 = oligophagy; 3 = polyphagy. The environmental index

was subdivided according to the environmental requirements of each species, and 4 groups were identified: 1 = ubiquitous; 2 = mesophilic; 3 = xerothermophilic; 4 = hygrophilic. The aim of the analysis was to describe relationships between biodiversity indicators and land-use indicators. Butterfly species richness and abundance, as well as the trophic index and the ecological index, were used as response variables in the models. Individual transects were classified into 4 classes of successional stages. Measures of advancement were the type of grassland (Table 1: Type) and form of use (Table 1: Description): 1—fallow land with a percentage of area without vegetation, mowed several times a year and biomass removed, treated as the youngest stage of succession; 2—fallow land/meadow mowed once a year with biomass not removed; 3—unmown fallow land/meadow with tall scrubs; and 4—unmown ecotones with tall scrubs and shrubs treated as the oldest stage of succession.

To investigate how butterfly species respond to land cover and management intensity, five land-use indicators were defined: Indicator 1 was defined as a percentage of land cover under particular land use (% area share). It was intended to reflect different management intensities of arable land and permanent grassland. Indicator 2 was related to a degree of ecological succession. This indicator was intended to reflect succession effects on biodiversity. Indicator 3 was related to the number of plant species per transect. Indicator 4 was related to the similarity of plant species (Bray–Curtis similarity) across the whole area, understood as species homogeneity. Indicator 5 was related to the number of melliferous plant species per transect.

ANOVA carried out with the use of IBM SPSS Statistics v. 26 was applied to compare multiple means. The test is designed to assess the significance of differences between multiple sample means for multiple groups [58]. To test whether food preferences (food index) and habitat preferences (habitat index) influence plot selection, an ANOVA analysis of variance was used. To test whether the stage of land succession and the form of land use differentiate the average number of butterflies, a Friedman test was conducted (the distribution of the average number of butterflies for all stages and the distribution of the sum of the number of butterflies for all forms deviated from the normal distribution). To test whether the stage of land succession differentiates biodiversity, a non-parametric Kruskal–Wallis test was conducted (the distribution of biodiversity for all forms deviated from the normal distribution).

With the purpose of examining the distribution of butterflies in the study transects and their response to the study area features, direct gradient analyses were carried out using the programs Canoco for Windows v. 4.56 and CanoDraw for Windows v. 4.14 [59,60]. First, a Detrended Canonical Correspondence Analysis (DCCA) was used to select the appropriate statistical model based on the longest gradient [61]. The data were not transformed. The mean Bray–Curtis plant similarity values for the study sites, the number of plant species, a degree of succession and % area shares were tested as environmental variables. With regard to butterfly distribution, the analyzed data showed a gradient of 1.2 SD units, so Redundancy Analysis (RDA) was used.

We then examined whether the different environmental variants affected the distribution of males and females and whether the sex of the species preferred the same habitats. With reference to the distribution of butterflies by sex, the data showed a gradient of 1.1 SD units, so the Redundancy Analysis (RDA) linear method was applied. Seven species with clear sexual dimorphism were used for this analysis. For each of the species, both numbers of males and females were included.

## 4. Results

As part of the study on the fauna of butterflies, 4212 individuals of 30 butterfly species were collected in 2020–2022 (Table 2). The most numerous was the family *Nymphalidae*, comprising 15 species. Next, in abundance, were: the *Pieridae*, represented by six species, and the *Hesperiidae* and the *Lycaenidae*—each with five species. The highest population numbers were recorded for *Lycaena viregaureae* and *Maniola jurtina*, and then *Coenonympha pamphilus* and *L. tityrus*. High variability was observed between transects in terms of the

number of butterfly species and specimen abundance. The highest number of species (24) was found on the transect established on the mown post-agricultural land with biomass not removed (Transect no. 2) and in the ecotone forest–fallow (L) (20 species). The lowest species numbers were found on the transect established on unmown meadow (NK) and unmown post-agricultural land (Transect no. 5), 12 and 13, respectively. The transect established on the mown post-agricultural land with biomass not removed (2) was characterized by the highest abundance of butterfly specimens. The next most-abundant transect in terms of butterfly specimens was the mown fallow with removed biomass and the forest ecotone—fallow (L). The transect least visited by butterflies was the unmown meadow. Differences can also be observed in the number of transects where butterfly species occurred (Table 2). Species with 100% frequency (Fi) on all the transects were *Pieris rapae, Lycaena viragaureae, L. tityrus, Issoria lathonia, Melanargia galathea, Maniola jurtina, T. sylvestris* and *Coenonympha pamphillus*.

**Table 2.** The list of recorded butterfly species, including the numbers of specimens collected per transect and species frequency (Fi), in 2020–2022.

| Butterfly Species | Transect | | | | | | | Sum | Fi |
|---|---|---|---|---|---|---|---|---|---|
| | 2 | 3 | 5 | L | W | KZ | NK | | |
| *Aporia crataegi (Linnaeus, 1758)* | 57 | 6 | 0 | 15 | 0 | 3 | 3 | 84 | 71.43 |
| *Pieris brassicae (Linnaeus, 1758)* | 0 | 0 | 0 | 0 | 6 | 0 | 0 | 6 | 14.29 |
| *P.rapae (Linnaeus, 1758)* | 12 | 12 | 6 | 15 | 15 | 12 | 3 | 75 | 100 |
| *P. napi (Linnaeus, 1758)* | 0 | 0 | 0 | 0 | 0 | 3 | 0 | 3 | 14.29 |
| *Gonepteryx rhamni (Linnaeus, 1758)* | 21 | 24 | 0 | 3 | 3 | 0 | 3 | 54 | 71.43 |
| *Anthocharis cardamines (Linnaeus, 1758)* | 0 | 0 | 0 | 6 | 0 | 0 | 0 | 6 | 14.29 |
| *Lycaena phlaeas (Linnaeus, 1761)* | 12 | 9 | 6 | 24 | 0 | 3 | 0 | 54 | 71.43 |
| *L. virgaureae (Linnaeus, 1758)* | 222 | 228 | 72 | 141 | 24 | 15 | 6 | 708 | 100 |
| *L.tityrus (Poda, 1761)* | 144 | 99 | 9 | 90 | 21 | 54 | 18 | 435 | 100 |
| *Cyaniris semiargus (Rottemburg, 1775)* | 0 | 0 | 0 | 0 | 9 | 0 | 0 | 9 | 14.29 |
| *Polyommatus icarus (Rottemburg, 1775)* | 63 | 96 | 6 | 15 | 0 | 9 | 0 | 189 | 71.43 |
| *Nymphalis antiopa (Linnaeus, 1758)* | 3 | 3 | 0 | 0 | 0 | 0 | 0 | 6 | 28.57 |
| *Inachis io (Linnaeus, 1758)* | 0 | 6 | 0 | 0 | 6 | 0 | 0 | 12 | 28.57 |
| *Vanessa atalantha (Linnaeus, 1758)* | 3 | 3 | 0 | 0 | 0 | 0 | 0 | 6 | 28.57 |
| *Vanessa cardui (Linnaeus, 1758)* | 0 | 3 | 0 | 0 | 0 | 0 | 0 | 3 | 14.29 |
| *Aglais urticae (Linnaeus, 1758)* | 9 | 18 | 3 | 0 | 12 | 6 | 0 | 48 | 71.43 |
| *Issoria lathonia (Linnaeus, 1758)* | 48 | 192 | 15 | 78 | 12 | 15 | 9 | 369 | 100 |
| *Melanargia galathea (Linnaeus, 1758)* | 138 | 12 | 9 | 24 | 24 | 60 | 18 | 285 | 100 |
| *Maniola jurtina (Linnaeus, 1758)* | 222 | 180 | 54 | 195 | 9 | 39 | 9 | 708 | 100 |
| *Aphantopus hyperanthus (Linnaeus 1758)* | 24 | 0 | 0 | 21 | 3 | 0 | 0 | 48 | 42.86 |
| *Coenonympha pamphilus (Linnaeus, 1758)* | 369 | 48 | 6 | 135 | 18 | 42 | 15 | 633 | 100 |
| *C. glycerion (Borkhausen, 1788)* | 15 | 0 | 9 | 3 | 3 | 6 | 12 | 48 | 85.71 |
| *Argynnis adippe (Denis and Schiffermüller, 1775)* | 18 | 6 | 0 | 3 | 0 | 0 | 0 | 27 | 42.86 |
| *Argynnis aglaja (Linnaeus, 1758)* | 3 | 0 | 3 | 9 | 0 | 0 | 0 | 15 | 42.86 |
| *Melitaea cinxia (Linnaeus, 1758)* | 21 | 0 | 0 | 33 | 3 | 0 | 0 | 57 | 42.86 |
| *Polygonia c-album (Linnaeus, 1758)* | 0 | 0 | 0 | 3 | 0 | 0 | 0 | 3 | 14.29 |
| *Thymelicus lineola (Ochsenheimer, 1808)* | 9 | 0 | 0 | 15 | 30 | 15 | 6 | 75 | 71.43 |
| *T. sylvestris (Poda, 1761)* | 135 | 9 | 33 | 21 | 12 | 24 | 6 | 240 | 100 |
| *Ochlodes sylvanus (Esper, 1777)* | 3 | 0 | 0 | 0 | 0 | 0 | 0 | 3 | 14.29 |
| *Hesperia comma (Linnaeus, 1758)* | 0 | 3 | 0 | 0 | 0 | 0 | 0 | 3 | 14.29 |
| Number of specimens | 1551 | 957 | 231 | 849 | 210 | 306 | 108 | 4212 | |
| Number of species | 22 | 19 | 13 | 20 | 17 | 15 | 12 | 30 | |

The highest Shannon–Wiener Diversity Index H characterized the transect established on mown fallow without harvested biomass and mown fallow with harvested biomass and the fallow–forest ecotone. The remaining transects were characterized by lower values of the diversity index (Table 3).

Based on the collected material, the dominance structure of butterfly species was determined [22]. The analysis showed a large number of species with low abundance and a small number of species with high abundance (Table 4). The abundance of recedents and sub-recedents in the study area was characteristic of well-developed ecosystems, as a large number of species with low abundance indicates the intrinsic diversity of environments [62].

This regularity can be seen in the case of the dominance of individual species in the area "Krzywda" (Table 4).

**Table 3.** The number of butterfly species, specimens and the Shannon–Wiener Diversity Index (H) in 2020–2022.

|  | 2 | 3 | 5 | L | W | KZ | NK |
|---|---|---|---|---|---|---|---|
| the number of specimens | 1551 | 957 | 231 | 849 | 210 | 306 | 108 |
| the number of species | 22 | 19 | 13 | 20 | 17 | 15 | 12 |
| H | −0.538 | −0.356 | −0.118 | −0.345 | −0.121 | −0.156 | −0.067 |

**Table 4.** Butterfly dominance structure in 2020–2022.

| Transect | eR I | R II | sR III | sD IV | D V | eDVI |
|---|---|---|---|---|---|---|
| 2 | 6 | 7 | 3 | 6 | 0 | 0 |
| 3 | 7 | 6 | 3 | 3 | 0 | 0 |
| 5 | 9 | 1 | 3 | 0 | 0 | 0 |
| L | 6 | 9 | 2 | 3 | 0 | 0 |
| W | 8 | 9 | 0 | 0 | 0 | 0 |
| KZ | 6 | 5 | 4 | 0 | 0 | 0 |
| NK | 8 | 4 | 0 | 0 | 0 | 0 |
| average number of species | 7.14 | 5.86 | 2.14 | 1.71 | 0 | 0 |

The list of collected butterfly species, including information on preferred habitats and host plants, their ecological index as well as trophic index is presented in Table 5. In the group of monophagous butterfly species, there were on average 57.6 specimens/transect/year. In the case of 10 monophagous species present on the study area, the highest average specimen numbers were found on the transects: 3 (18.8), 2 (16.5) and L (11.6). The lowest average specimen numbers were observed on the transects: NK (1.2), W(2.8), KZ (3.2) and 5 (3.5). The group of oligophagous species included 13 specimens, with an average of 26.5 specimens/transect/year. The highest number of oligophages was caught on transect 2 (10.9) and the lowest on NK (0.9). On the other transects, oligophagous specimen numbers/year was as follows: 3 (4.1), 5 (1.4), L (3.6), W (2.2), KZ (3.1).

**Table 5.** The list of collected butterfly species, including information on preferred habitats and host plants, their ecological index (Ie: u—ubiquitous species; m—mesophilic species; x—xerothermophilic species and h—hygrophilic species) as well as trophic index (It: m—monophagus; o—oligophagus and p—polyphagus). Habitat preferences were defined based on the literature [19,25–57].

| Butterfly Species | Habitat | Host Plant | Ie | It | Literature |
|---|---|---|---|---|---|
| *Aporia crataegi* (L.) | Deciduous forests, ruderal areas, meadows, orchards, agricultural fields | *Family Rosaceae: Crataegus monogyna,Prunus, Sorbus,Frangula* | m | m | [27] |
| *Pieris brassicae* (L.) | Ruderal areas, forest edges, scrub vegetation, roadside vegetation, dry and wet meadows, mixed forests | *Family Brassicaceae: Brassicae, Sinapsis arvensis, Raphanus raphanistrum, Tropaeolum majus* | u | p | [28] |
| *P.rapae* (L.) | Ruderal areas, forest edges, roadside vegetation, scrub vegetation, dry and wet meadows, gardens and recreational areas, parks | *Family Brassicaceae: Brassicae, Lepidium, Arabis, Alliaria petiolata* | u | o | [28,29] |

**Table 5.** *Cont.*

| Butterfly Species | Habitat | Host Plant | Ie | It | Literature |
|---|---|---|---|---|---|
| *P. napi* (L.) | Forest edges, ruderal areas, scrub vegetation, mixed forests, roadside vegetation, dry and wet meadows, gardens and recreational areas, parks | *Thlaspi arvense, Thlaspi alpestre, Alliaria petiolata, Brassica campestris, Brassica rapa, Brassica napus, Brassica oleracea, Raphanus raphanistrum, Raphanus sativus, Armoracia rusticana, Rorippa islandica, Cardamine amara, Arabis alpina, Hesperis matronalis, Berteroa incana, Reseda odorata, Tropaeolum majus, Calendula officinalis, Cardamine leucantha, Cardamine niponica, Rorippa isbandica, Rorippa sylvestris* [3]. | u | o | [28] |
| *Gonepteryx rhamni* (L.) | Forest edges, mixed forests ruderal areas, scrub vegetation, dry and wet meadows, roadside vegetation, gardens and recreational areas | *Family Rhamnaceae: Rhamnus cathartica, Frangula alnus* | m | o | [30] |
| *Anthocharis cardamines* (L.) | Forest edges, forest roads, forest glades and cutting areas, wet meadows, scrub vegetation, roadside vegetation, parks | *Family Brassicaceae: Cardamine pratensis* L., *Alliaria petiolata, Arabis glabra* (L.) Bernh., *Sisymbrium officinale* (L.) Scop. | h | o | [31,32] |
| *Lycaena phlaeas* (L.) | Field borders, fallow lands, forest glades | *Rumex acetosella* L., *Rumex acetosa* L. *Polygonum* L. | m | m | [33] |
| *L. virgaureae* (L.) | Forest edges, scrub vegetation, fallow lands and natural areas, dry and wet meadows, roadside vegetation | *Rumex* spp. | m | m | [34,35] |
| *L.tityrus (Poda)* | Forest edges, scrub vegetation, roadside vegetation, dry and wet meadows, ruderal areas | *Rumex acetosella, Rumex acetosa* L., | m | m | [33] |
| *Cyaniris semiargus (Rott.)* | Forest edges, ruderal areas, dry and wet meadows, scrub vegetation, forest glades, fallow lands | *Trifolium pratense* L., *Trifolium medium* (L.) *Lasius niger* | m | m | [36,37] |
| *Polyommatus icarus (Rott.)* | Forest edges, ruderal areas, roadside vegetation, dry and wet meadows, forest glades, fallow lands | *Family Fabaceae Trifolium pratense* L., *Trifolium medium* (L.), *Medicago* L. *Lotus* L., *Ononis* L. | m | o | [38,39] |
| *Nymphalis antiopa* (L.) | Mixed forests, forest edges, forest pathways; sometimes: meadows and scrub vegetation | *Salix* spp., *Populus* spp., *Betula* spp., *Ulmus* spp. | m | o | [40,41] |
| *Inachis io* (L.) | Forest edges and glades, gardens, orchards, wastelands, pastures | *Urtica dioica* L. | u | m | [42–44] |
| *Vanessa atalanta* (L.) | Forest edges, dry and wet meadows, fallow lands, ruderal areas, roadside vegetation, scrub vegetation, deciduous forests, parks, gardens | *Urtica dioica* L. | u | m | [45] |

**Table 5.** *Cont.*

| Butterfly Species | Habitat | Host Plant | Ie | It | Literature |
|---|---|---|---|---|---|
| *Vanessa cardui* (L.) | Ruderal areas, parks, field borders, fallow lands, roadside vegetation | *Cirsium Mill.*, *Urtica dioica* L., *Carduus* L., *Onopordum* L. | m | p | [44,46] |
| *Aglais urticae* (L.) | Sunny forest glades, meadows, gardens, ruderal areas | *Urtica dioica* L. | x | m | [44] |
| *Issoria lathonia* (L.) | Dry or sandy habitats, fallow lands, ruderal areas, roadside vegetation | *Viola arvensis* | m | m | [37] |
| *Melanargia galathea* (L.) | Forest edges, ruderal areas, scrub vegetation, dry and moderately wet meadows, railway embankments | *Family Poaceae, Gramineae* | m | o | [47] |
| *Maniola jurtina* (L.) | Ruderal areas, scrub vegetation, dry and wet meadows, forest edges, roadside vegetation, railway embankments | *Lolium perenne, Festuca rubra* L., *Poa pratensis* L. | m | p | [34] |
| *Aphantopus hyperantus* (L.) | Ruderal areas, scrub vegetation, forest grasslands, dry and wet meadows, roadside vegetation, roadside vegetation, railway embankments | *Brachypodium, Dactylis, Festuca* L., *Bromus, Poa pratensis, Carex, Agrostis, Holcus* | m | p | [48] |
| *Coenonympha pamphilus* (L.) | Forest edges, ruderal areas, scrub vegetation, dry and wet meadows, roadside vegetation | *Family Poaceae: Festuca* spp., *Poa* spp., *Agrotis* spp. | u | p | [33,48] |
| *C. glycerion (Borkh.)* | Forest edges, ruderal areas, scrub vegetation, dry and wet meadows, roadside vegetation, grasslands with trees and shrubs | *Molinia* L., *Festuca* L. | m | p | [35,49] |
| *Argynnis adippe (D and S)* | Sunny forest glades, coppices, clearcutting areas, railway embankments | *Viola canina* L., *Viola odorata* L., *Viola hirta* L. | m | o | [35,50] |
| *A. aglaja* (L.) | Wild-flower meadows, scrub vegetation, peat meadows, railway embankments, forest edges, fallow lands and ruderal areas, dry and wet meadows | *Family Violaceae* | m | o | [35] |
| *Melitaea cinxia* (L.) | Forest glades, extensively used pastures, wild-flower meadows | *Plantago* L. | m | o | [51,52] |
| *Polygonia c-album* (L.) | Ruderal areas, gardens, forest glades and roads | *Urtica dioica, Humulus lupulus, Ulmus glabra, Salix caprea* L. | m | p | [53,54] |
| *Thymelicus lineola (Ochs.)* | Meadows, gardens and parks, forest glades and edges | *Family Poaceae* | m | o | [35,55] |
| *T. sylvestris (Poda)* | Forest edges, ruderal areas, scrub vegetation, dry and wet meadows, roadside vegetation | *Holcus, Dactylis glomerata* | m | o | [55] |

**Table 5.** *Cont.*

| Butterfly Species | Habitat | Host Plant | Ie | It | Literature |
|---|---|---|---|---|---|
| *Ochlodes Sylvanus (Esper)* | Meadows, forest glades, forests | *Family Poaceae* | m | o | [56] |
| *Hesperia comma* (L.) | Xerothermic grasslands, fallow lands, forest edges and glades | *Festuca ovina, Lolium perenne* L., *Corynephorus canescens* L. | m | m | [57] |

The results of the analyses showed that in the case of seven polyphagous species, on average, 69 specimens were found per transect/year. The highest numbers were found on the transects 2 (30) and L (17), and the lowest on NK (1.7) and W (1.8). On the transects 3, 5 and KZ, we observed 11, 3.2 and 4.1 specimens, respectively. On the basis of the results of ANOVA, it can be concluded that, at a probability level of $p < 0.05$, that differences in land use had an effect on the occurrence of butterflies on transects 2, 3, 5 and L, which means that the food preferences of butterflies influenced the selection of these particular transects. On the other hand, in the remaining transects W, KZ and NK, at $p > 0.05$, we rejected this hypothesis and concluded that food preferences probably had no effect on butterfly occurrence.

The analysis of the ecological index showed the preference for habitat in butterflies. Of six ubiquitous species, on average, 40.8 specimens were caught per transect/year. The highest numbers were observed on transect 2 (on average 21.3 specimens), and the lowest on transects 5 and NK, with average numbers 0.6 and 1, respectively. The numbers of ubiquitous butterfly species on other transects under the study were: 3 (3.8), L (8.3), W (2.5), KZ (31). In the case of 22 mesophilic species, on average, 51.8 specimens per year were collected. Most specimens were observed on transect 2 (average 17.5) and transect 3 (average 13). The lowest average specimen numbers were found on the following transects: NK—1.3 and W—2.3. On other studied transects, the average specimen numbers were: L—10.5, 5—3.2 and KZ—3.6. The group of xerothermic and hygrophilous species included one species each. Of 16 specimens of xerothermic species, the highest average numbers were found on transects 3 and W: 6 and 4, respectively. One hygrophilous species was found only on transect L. Based on an ANOVA analysis of the obtained results regarding the ecological index, at $p < 0.05$, we showed that the habitat conditions influenced the distribution of butterflies.

The Friedman test conducted to determine whether the stage of site succession differentiates the average number of butterflies showed significant differences, $\chi^2(3, N = 34) = 10.63$, $p < 0.05$. Post hoc comparisons were conducted. The mean number of butterflies for stage 1 (mean rank = 2.66) was significantly higher than for stage 4 (mean rank = 1.97), $p < 0.05$. The mean number of butterflies for stage 2 (mean rank = 2.79) was significantly higher than for stage 4, $p < 0.05$. In contrast, whether the stage of land succession differentiates biodiversity differences did not prove to be significant, $\chi^2(3, N = 42) = 1.64$, $p > 0.05$.

Upon analyzing whether the form of land use differentiates the quantitative occurrence of butterflies, the differences proved to be significant, $\chi^2(6, N = 34) = 50.99$, $p < 0.001$. Post hoc comparisons were conducted. Butterfly abundance on transect NK (mean rank = 2.84) was significantly lower than on transect 2 (mean rank = 5.29), significantly lower than on transect 3 (mean rank = 4.57), significantly lower than on transect 5 (mean rank = 3.18) and significantly lower than on transect L (mean rank = 4.69). Butterfly abundance on transect 2 was significantly higher than on transect KZ (mean rank = 3.63) and significantly higher than on transect 5. The analysis showed that the form of land use differentiates the quantitative occurrence of butterflies. The analysis showed that the stage of succession differentiates the quantitative occurrence of butterflies.

The results of the RDA with all species (Figure 2) show that the number of plant species in the transects as well as their similarity indicate a negative correlation with the stage of succession and the % share of area. As the number of melliferous plant species and

number of plant species increases, the species similarity between the transects, i.e., their homogeneity, increases. In contrast, the older the stage of succession and the % share of area, the fewer the number of plant species and melliferous plant species, and the lower their similarity. Transects 2 and L show a positive correlation with the number of plant species as well as their similarity. A positive correlation with plant similarity is also shown by transect 3. The transects positively correlated with the degree of succession and the % share of area are: NK, W and 5. The transect KZ shows no significant correlation. All of the 15 butterfly species analyzed are placed on the right side of the diagram. This analysis indicates that they are positively correlated with vegetation and its similarity, and negatively correlated with the succession stage and % area share. The exceptions to these patterns are two species *Vanessa cardui* and *Hesperia comma*, which both show an increase with an increasing value of % share of area. In addition, these species show a negative correlation with species *Coenonympha pamphilus*, *Aphantopus hyperantus*, *Aporia crataegii* and *Melitaea cinxia*. The remaining species tested show positive correlations with each other. This analysis showed that the degree of succession as well as the similarity of plant species (homogeneity of the area) determined the occurrence of butterflies along the more important first axis.

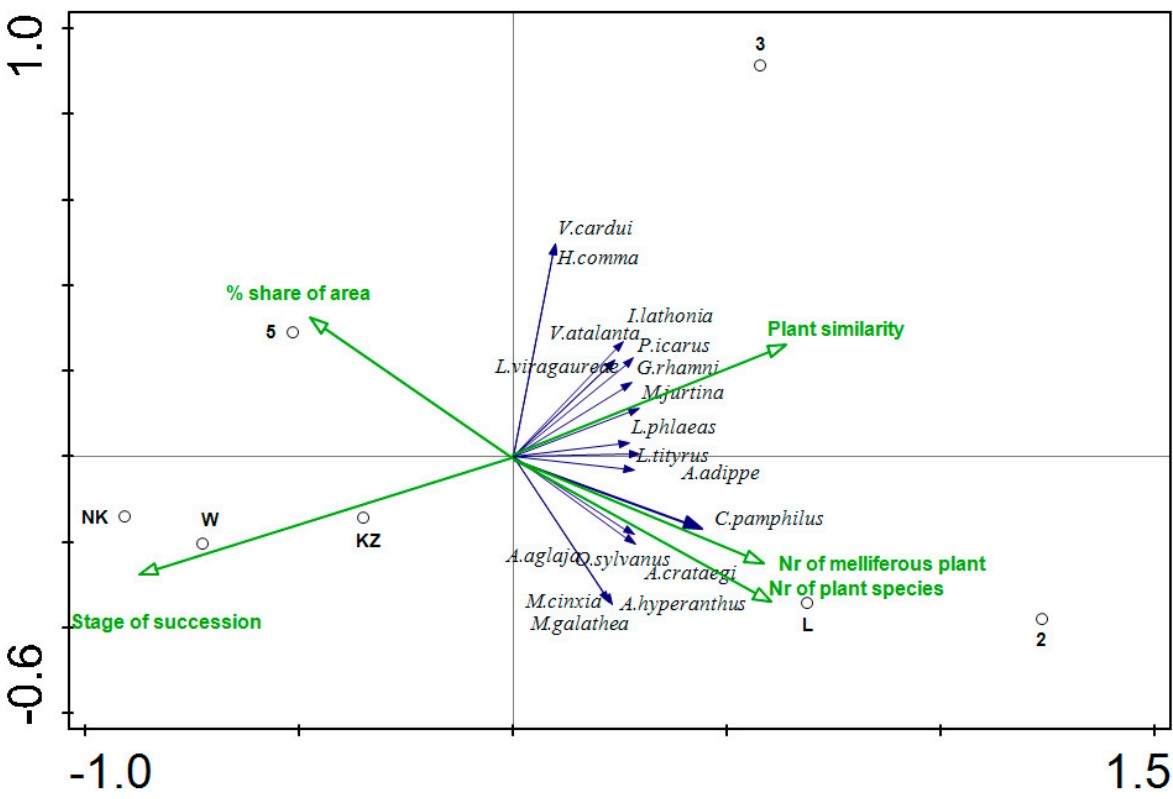

**Figure 2.** Redundancy Analysis (RDA) for transects (black circles), and 15 butterfly species showing the highest correlation (dark blue arrows) and environmental variables (green arrows). The 15 species with the best fit into the ordination space are shown.

Subsequently, the occurrence of the sexes of individual species in relation to each other and environmental variables was examined (Figure 3). A positive correlation is found between males and females in the species L. tityrus, Maniola jurtina, Polyommatus icarus, Aphantopus hyperantus and Coenonympha glicerion, whereas a negative correlation—in Genopterix rhamni. A zero correlation is shown by Lycaena virgaurea. The number of females of Polyommatus icarus, Lycaena virgaurea, L. tityrus increases with the increasing number of plant species and similarity. Their preferred areas are L, 2 and 3. Males of the species Lycaena virgaurea, Aphantopus hyperantus, Coenonympha glicerion showed a positive correlation with the number of plant species and a negative correlation with %

share of area. Coenonympha glicerion is the only species who show in the case of males a positive correlation with a degree of succession.

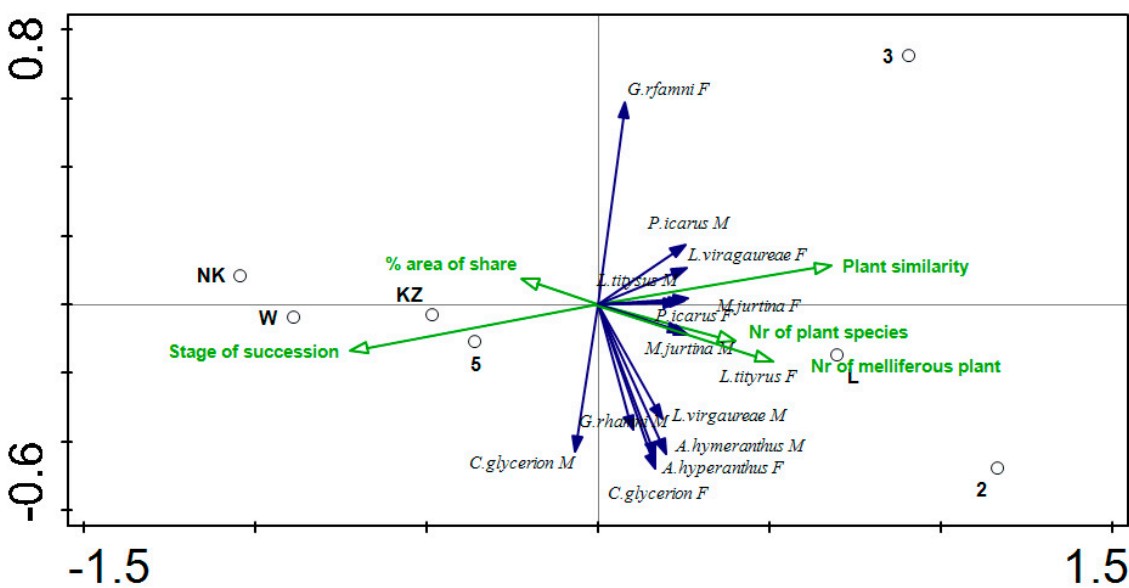

**Figure 3.** Redundancy Analysis (RDA)for transects (black circles), and 7 butterfly species with clear sexual dimorphism (dark blue arrows) and environmental variables (green arrows). M—male, F—female.

## 5. Discussion

The main natural factors shaping the population dynamics of butterflies primarily include habitat preference and host plant occurrence. To this, a whole range of anthropogenic factors that overlap with or complement or modify natural factors must be added. The anthropogenic factor with adverse effects is habitat loss due to physical destruction, fragmentation or change of land use [12,63–65]. On the other hand, humans, through various land uses, can create environments that foster the occurrence of specific species, as well as shape the composition and abundance of the species [3,5,64,65]. A wide range of factors shaping changes in the abundance of butterflies makes these insects sensitive indicators of the changes taking place [66]. As a part of the study on the fauna of butterflies, 4212 individuals of 30 butterfly species were collected, which represented approximately 18% of all the butterfly species that inhabit Poland [67,68]. The study area showed differences in the species composition and population numbers of butterflies across transects varying in land use. This most likely means that the simultaneous population growth of some butterfly species is fostered by a particular form of economic use, while another may be detrimental [69]. The various species under the study were unevenly represented. Both dominant, co-dominant and sporadic species could be distinguished. The uneven proportions of individual butterfly species found in the study area represents a typical dominance structure. This means that within the "Krzywda" study area, well-structured ecosystems are formed with the characteristic small number of species and a high abundance of specimens, along with a large number of species with low abundance of specimens.

The species composition shows the dominance of butterflies commonly found in Poland, adapted to live in moderately humid environments, trophically related to the vegetation of meadows, cultivated fields, fallow lands and mid-field baulks [5,25], which are the most favorable and stable sites for their development in the studied area. Undoubtedly, human economic activities of mowing and cutting down trees and shrub undergrowth have contributed to the distribution and abundance of the butterflies observed. The results of the analyses showed the large number of species with low abundance and the small number of species with high abundance. The abundance of recedents and subrecedents in the study area was characteristic of properly developed ecosystems, as the high number

of low-abundance species is indicative of the intrinsic diversity of the environment [62]. This regularity can be observed in the case of the dominance of individual species in the "Krzywda" area. This is most likely a result of the mosaic pattern of the study area and the presence of microhabitats [22,68–70], i.e., habitat diversity [71–74]. Butterflies are sensitive to habitat quality and management; therefore, they are a good indicator for assessing the impact of human activities [75]. Butterflies are relatively mobile and many species—especially those more in demand and are often threatened—require several habitats to complete their life cycle [76]. This study showed that the environmental requirements of butterfly individual species influence their distribution. The diverse management of the "Krzywda" landscape favored the abundance of mesophilic and ubiquitous species, whereas xerothermophilic and hygrophilous habitats were not conducive to butterfly abundance. Once divided by habitat requirements, butterflies were most abundant in mown fallow areas with or without harvested biomass and in the ecotone forest–fallow. In addition to habitat factors, the choice of sites was also determined by food preferences. Polyphagous species chose mown fallows without harvested biomass as well as forested areas. The same was true for monophagous species, except that they were additionally abundant in mown fallows with harvested biomass. Specimens of oligophagous species were most likely to be found in mown fallows without harvested biomass. Previous studies in the area have shown differences in soil and plant characteristics among a number of selected ecosystems and ecotones, as well as differences in butterfly communities. They demonstrated that butterflies showed different responses to different features of the various sites studied, which can be explained by differences in relation to their ecological characteristics, such as feeding preferences [5]. Unmown meadows, the marsh ecotone–fallow as well as unmown fallow land were not preferred by butterflies. The ANOVA showed that the presence of butterfly specimens in these areas was most likely not due to food preferences. Szabó et al. [77], analyzing overwintering, flight period, wing length, diet and territorial behavior, showed that habitat type had the strongest filtering effect on butterfly functional traits of the three design variables: habitat type, management and landscape context. They also indicated that habitat type had the strongest influence on host plant specificity. The orchard meadows they studied provided habitat for generalist species feeding on a wider range of host plants, i.e., polygamous, as butterfly species with narrow specializations are usually found in calcareous grasslands. They concluded, in agreement with Betzholtz et al. [78], that this may be due to the fact that calcareous grasslands are characterized by a much more diverse herbaceous vegetation, and specialized species displace generalist species into lower-quality habitats.

The distribution of butterflies analyzed in relation to the number of plant species allowed for the conclusion that, with a high probability, it influenced the abundance of butterflies. The transects established on mown fallow lands with harvested biomass and with unharvested biomass, as well as in the forest ecotone–fallow, were characterized by the highest butterfly abundance. At the same time, the most plant species were recorded in these transects. It can therefore be concluded that plant species richness influenced the abundance of butterflies. This has been confirmed by other studies showing that different forms of land use, as well as plant species richness resulting from differentiated landscape management, influenced butterfly occurrence [5,12,65,69,70,79,80]. Plant species richness enables the availability of more host plants and food for foraging species [65,66,81,82]. The analysis also showed that the younger the ecological succession stages, the higher the species richness of butterflies, whereas the size of the area under a given land use did not influence the observed increased abundance of butterflies. Undoubtedly, the majority of butterfly species occurring in Poland prefer open areas classified as early succession stages—thus, the maintenance of such stages is necessary [8]. This study showed sex preferences, as well as preferences for plant species richness or the succession stage. This may be due to gender preference for flowers with different nectar compositions [83], sex-specific traits, or intraspecific variation [84–86]. Perhaps, due to high densities, females

seeking to maximize a survival rate of eggs laid wish to reduce intraspecific competition between offspring and move to the areas with high plant species richness.

The results presented show that many factors can affect the occurrence of butterflies. In order to determine which one had the most significant effect on their distribution in the study area, a multivariate statistical analysis of Canoco 5 was applied. The influence of five factors that could be recorded in numerical form, i.e., stage of succession, number of species of melliferous plants, number of plant species, area size as well as plant similarity (habitat homogeneity), was analyzed. Our study showed that the stage of succession as well as plant species similarity determined the occurrence of butterflies along the more important first axis. The younger the stage of succession and the greater the species homogeneity, the more the number of butterfly species and their individuals. The degree of succession in the area is strongly correlated with the form of land use, which is the result of human economic activity. The area is shaped by the hand of humans, which therefore directly or indirectly affects the characteristics of habitats. Therefore, it can be concluded that humans shaping the size of land use patterns and spatial arrangement contribute to the distribution of butterflies in the area.

A fundamental question regarding the practical aspects of conservation is to what extent the type of management of ecosystems affects their characteristics and biodiversity. A study in the Italian Alps [79] showed that agricultural management, plant species richness and landscape diversity had a significantly positive effect on the species richness of butterflies in a grassland–forest mosaic. Swengel [80], studying grass prairie and pine barrens, found that for the conservation of specialized butterfly species, the consistency of management within a site along with diversity between sites was desirable. And Morris [87], studying grasslands, stressed the importance of integrating theoretical and experimental aspects of grassland ecology with the practical knowledge of reserve managers and conservation officers.

The spatial scale is also undoubtedly important. A study by Aviron et al. [3] in Switzerland, on the impact of agri-environmental measures on butterflies using a multi-scale approach, found that the effectiveness of ecological compensation areas depends on both local site conditions and the number of ecological compensation areas and semi-natural elements in the surrounding landscape. In contrast, Scheper et al. [88] in their meta-analysis showed that agri-environmental programs can create a contrast in floral resources that affect the response of pollinators as butterflies, and this response is moderated by the landscape context and type of agricultural land. In addition, Szyszko et al. emphasized that some species require different stages of succession or ecosystems in the broader landscape to establish their populations [89]. Previous studies [5] in the area have shown different responses of butterfly and runner groupings to different features of the different sites studied, which can be explained by differences in relation to their ecological characteristics, such as feeding or habitat preferences. We concluded that management practices in agricultural and forest ecosystems have a significant impact on the formation of beetle and butterfly assemblages. Large-scale management strategies are needed to effectively protect species diversity. Therefore, large-scale management strategies are needed to preserve or create landscapes of high conservation value to protect such species. At the landscape scale, the number of habitats, as well as their isolation, are also important factors affecting the presence and distribution of butterfly species [90,91]. For this reason, further research on the impact of human activities on this group of animals is important.

## 6. Conclusions

The butterflies showed different responses to the different characteristics of the various sites studied, which can be explained by differences in relation to their ecological characteristics, such as food or habitat preferences. We concluded that management practices in agricultural and forest ecosystems, such as those we studied, have a significant impact on the formation of butterfly communities. The observations to date seem to warrant the conclusion that species diversity and the abundance of butterflies can be influenced by

deliberate human activity in a given area. Therefore, it can be concluded that it is possible to control population dynamics of butterflies through the changes in ecological succession stimulated by human economic activities. Diverse landscape management will allow us to provide sites that meet the needs of butterflies for larval development and shelter, as well as to keep the mosaic of habitats and their surroundings in good condition so that the ecosystem functions and resources needed by butterfly species are maintained. Large-scale management strategies are needed to effectively protect species diversity. Developed environmental engineering means that humans can interfere with the species composition of a given ecosystem to achieve a desired effect. However, human activity will be successful only when the needs of individual species in a given area are appropriately understood.

**Author Contributions:** Conceptualization: K.S.-P.; methodology: K.S.-P. and I.D.; validation: K.S.-P. and I.D.; investigation: K.S.-P. and I.D.; resources: K.S.-P., I.D. and M.K.; data curation: K.S.-P. writing—original draft preparation: K.S.-P.; writing—review and editing: K.S.-P., I.D. and M.K.; visualization: K.S.-P. All authors have read and agreed to the published version of the manuscript.

**Funding:** This research received no external funding.

**Institutional Review Board Statement:** Not applicable.

**Informed Consent Statement:** Not applicable.

**Data Availability Statement:** The data belong to the Institute of Environmental Protection—National Research Institute, Warsaw. Data may be accessed by contacting the first author of the publication.

**Conflicts of Interest:** The authors declare no conflict of interest.

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
