# Peer review of "Diversity in Landscape Management Affects Butterfly Distribution"

_sustainability, doi:10.3390/su152014775_

Round 1
Reviewer 1 Report
1. The paper touches on an important issue. However, it is based on research carried out at only one field site. Consequently, its results and the conclusions drawn from them cannot be considered reliable.
2. As presented, the research is of a local nature and should not be published in “Sustainability” as it is outside the scope of the journal.
3. Moreover, the first two of the four research hypotheses are truisms and do not need to be proven. They are derived from the biology of each species and are obvious.
4. Most importantly, some of the analyses are based on the use of coefficients that are no longer in use for the determination of species diversity. The Shannon-Wiener index, for example, is not an index of the diversity or richness of species. In fact, because it is a probability/entropy index, it is considered obsolete because it is biologically meaningless (see: Jost, L. Entropy and diversity. Oikos 2006, 113, 363–375).
5. A significant proportion of the cited publications, which form the basis for the assignment of butterfly species to specific biotopes, are atlases or even textbooks.
6. In light of all of the above, I believe that the paper should be rejected.
I have detected only several minor typos.
Author Response
Response to Reviewer 1 Comments
Thank You very much for taking the time to review this manuscript. Thank You for your comments and remarks. Your comments are very valuable to me and my team. Below are responses to the comments, and in the attached manuscript, changes have been highlighted in red as suggested by all Reviewers. I hope that the corrections and explanations will make You find the article interesting and meet the requirements for publication in this journal.
With best regards
Authors
Comments 1,2 : The paper touches on an important issue. However, it is based on research carried out at only one field site. Consequently, its results and the conclusions drawn from them cannot be considered reliable. As presented, the research is of a local nature and should not be published in “Sustainability” as it is outside the scope of the journal.
Response 1,2: As described in the methodology of the presented manuscript, the research was carried out in one research facility called "Krzywda" with an area of over 160 ha, located in north-western Poland. This is an area that 30 years ago was designated exclusively for scientific research and was no longer used to generate income. This creates opportunities to maintain a permanent experimental pattern of land use and observe its impact on ecosystems.
The main thread of research undertaken here over 30 years ago is to observe the possibilities of controlling the succession process and, consequently, controlling the species diversity of groups of wild fauna, in particular entomofauna with a relatively low dispersal power (the area of the facility seems to be sufficient here).
In this respect, the facility is examined in many aspects, and its scientific usefulness is determined by the following factors: documented history of use of this area, going beyond the time frame of the research, exclusion from economic activity in the aspect of deriving financial benefits, which allows maintaining a constant factor, which is constant for individual research fields of the facility, form of land use (various forms of mowing with different frequency, with or without removing the mowing, artificial removal of saplings of trees and shrubs in forestless areas, artificial afforestation, cessation of use).
These land use/maintenance patterns, which have been constant for 30 years, allow for the observation of the managed development of the succession process and, consequently, changes or lack thereof in the species composition of entomofauna, in particular ground beetles (Carabidae, Col.) and, in recent years, also butterflies. Data on avifauna and plant species were also collected periodically. Selected soil parameters are also periodically tested.
Due to the above, it is difficult to find another, similar research facility in Poland that would enable research on the succession process, which requires constant, long-term observations. Due to the dispersal power of the observed groups of fauna species, its area seems to be sufficient.
Comments 3: Moreover, the first two of the four research hypotheses are truisms and do not need to be proven. They are derived from the biology of each species and are obvious.
Response 3: Thank You for pointing this out. We can't quite agree with this comment. There may be hypotheses formulated in general terms, but they refer specifically to the seven types of land management presented in the work (Table 1). The research aimed to demonstrate differences in the number and species diversity of butterflies depending on the land management method. It should be emphasized that in the past, all the studied areas were used for agriculture. Given the main themes of the research (explained in Response 1,2 above) undertaken at the "Krzywda" research site for 30 years (the possibility of controlling succession), perhaps the hypotheses presented are not a truism. The research points to the habitat preferences of butterflies not in natural, but man-made ecosystems.
Comments 4: Most importantly, some of the analyses are based on the use of coefficients that are no longer in use for the determination of species diversity. The Shannon-Wiener index, for example, is not an index of the diversity or richness of species. In fact, because it is a probability/entropy index, it is considered obsolete because it is biologically meaningless (see: Jost, L. Entropy and diversity. Oikos 2006, 113, 363–375).
Response 4: Thank you for pointing this out. I would like to point out the Shannon-Wiener index is still in use and can be found in various publications (for example: DOI: 10.1088/1755-1315/486/1/012081, 10.1016/j.biocon.2004.12.012, DOI: 10.1186/s41936-023-00327-9 )
Comments 5: A significant proportion of the cited publications, which form the basis for the assignment of butterfly species to specific biotopes, are atlases or even textbooks.
Response 5: Thank You for pointing this out. We agree with this comment. Therefore, this literature item has been removed and replaced with literature with references to studies that specifically address each butterfly species. (Table 5) page numer 8-11,19-20 and line 276,569,572,585, 600-670
Comments 6: In light of all of the above, I believe that the paper should be rejected.
Response 6 : We hope that our answers and corrections are sufficient
Reviewer 2 Report
The review comments and suggestions are as follows:
1. The paper explores an interesting and valuable question, which is how humans can design spatial patterns that better serve the survival of organisms, thus enhancing biodiversity and protecting ecosystems. However, there are some unclear aspects in the design of the study. For instance, while it is evident that different landscape patterns/land uses have an impact on butterfly populations, does this mean that the land use patterns, spatial layout, or landscape patterns in the study area are the primary reasons for the distribution characteristics of butterflies in the area? How does the author differentiate the impact of landscape patterns from other unknown factors on butterfly distribution? What are the degrees and mechanisms of influence of different factors? Furthermore, what are the uncertainty factors in this study? These questions need to be discussed and analyzed in depth within the paper.
2. The quality of the figures and the standardization of tables in the paper need further improvement to meet the quality requirements for publication. Additionally, it is recommended to use a formula editor to generate and number each formula in the paper.
3. It is suggested to make the conclusions in the paper more concise. Conclusions should summarize valuable insights derived from the research results, rather than reiterate the phenomena observed in the study. Furthermore, based on the results of this study, it is recommended to supplement information about landscape management measures aimed at improving ecological diversity in the study area, providing valuable decision support for local land use and landscape planning practices.
Author Response
Response to Reviewer 2 Comments
Thank You very much for taking the time to review this manuscript. Thank You for your comments and remarks. Your comments are very valuable to me and my team. Below are responses to the comments, and in the attached manuscript, changes have been highlighted in red as suggested by all Reviewers. I hope that the corrections and explanations will make You find the article interesting.
With best regards
Authors
Comments 1: The paper explores an interesting and valuable question, which is how humans can design spatial patterns that better serve the survival of organisms, thus enhancing biodiversity and protecting ecosystems. However, there are some unclear aspects in the design of the study. For instance, while it is evident that different landscape patterns/land uses have an impact on butterfly populations, does this mean that the land use patterns, spatial layout, or landscape patterns in the study area are the primary reasons for the distribution characteristics of butterflies in the area? How does the author differentiate the impact of landscape patterns from other unknown factors on butterfly distribution? What are the degrees and mechanisms of influence of different factors? Furthermore, what are the uncertainty factors in this study? These questions need to be discussed and analyzed in depth within the paper.
Response 1: Thank You for pointing this out. We agree with this comment. We have tried to supplement the manuscript with the issues you raised. The topic is undoubtedly very important and widely considered, so I hope that, in part, the corrections will be sufficient for You. Page numer 13 line 349-351, page numer 15 line 422-432, page numer 16 line 453-496.
Comments 2: The quality of the figures and the standardization of tables in the paper need further improvement to meet the quality requirements for publication. Additionally, it is recommended to use a formula editor to generate and number each formula in the paper.
Response 2: Thank you for pointing this out. I have tried to send photos in better quality, I hope they will be more visible after publication.
Comments 3: It is suggested to make the conclusions in the paper more concise. Conclusions should summarize valuable insights derived from the research results, rather than reiterate the phenomena observed in the study. Furthermore, based on the results of this study, it is recommended to supplement information about landscape management measures aimed at improving ecological diversity in the study area, providing valuable decision support for local land use and landscape planning practices..
Response 3: Thank You for pointing this out. We agree with this comment. Therefore, we have made corrections in the abstract and conclusion. I hope they are sufficien. Page numer 17 line 498-513.
Reviewer 3 Report
Dear authors, congratulations for your research, I appreciate your results. A well-structured paper. Overall, a complex work, the analysis involving a number of important and varied components for the final result. An interesting paper and a pertinent analysis on
landscape management in order to maintain the biodiversity of ecosystems.
The results obtained and presented in tables, figures and diagrams are quite relevant to the proposed research and analysis.
The conclusions are supported by the analyzes done, complementing the existing information in this field.
Good luck
Author Response
Dear Reviewer
Thank You very much for taking the time to review this manuscript. Thank You for your positive review of our manuscript. We did not expect such good feedback. It is our dream to receive just such responses. Your comments are very valuable to me and my team. And motivate us to continue our work.
With best regards
Authors
Reviewer 4 Report
The manuscript “Diversity in landscape management affects butterfly distribution” (sustainability-2614030) describes the effect of different landscape habitats on butterfly diversity and abundance. Overall, I think the manuscript suits the journal and the research is well-conducted, but some clarifications are necessary.
Abstract
The abstract is too general. Some specific results regarding butterfly species and habitats should be included in the abstract.
Introduction
Lines 73-80: Only hypotheses that were actually tested should be included. For example, the hypothesis regarding quantitative occurrence of melliferous vegetation was not tested. Melliferous vegetation is not synonym of number of plant species.
Materials and Methods
Line 122: What do you mean by “phytosociological”?
Lines 126-131: the explanation of how plant cover was determined is confusing. A better explanation should be given.
Results
Table1. Define in the Materials and Methods how you determined the stage of succession.
Table 5. Host plant Crataegus L? Where did authors obtain the host plant information from? I would suggest adding a column with references of studies that specifically refer to each of the butterfly species and host plants mentioned. For example, for Aporia crataegi L., Jugovic et al. (2017) https://doi.org/10.1007/s10841-017-9977-z, for Pieris rapae L., Badenes-Pérez (2023), https://doi.org/10.3390/plants12112148
Some of the scientific names of butterfly species in Table 5 have missing parts. The full species name should be shown.
Author Response
Response to Reviewer 4 Comments
Thank You very much for taking the time to review this manuscript. Thank You for your comments and remarks. Your comments are very valuable to me and my team. Below are responses to the comments, and in the attached manuscript, changes have been highlighted in red as suggested by all Reviewers. I hope that the corrections and explanations will make You find the article interesting.
With best regards
Authors
Comments 1: The manuscript “Diversity in landscape management affects butterfly distribution” (sustainability-2614030) describes the effect of different landscape habitats on butterfly diversity and abundance. Overall, I think the manuscript suits the journal and the research is well-conducted, but some clarifications are necessary.
Response 1: Thank you very much for your opinion
Comments 2: Abstract. The abstract is too general. Some specific results regarding butterfly species and habitats should be included in the abstract.
Response 2: We agree with this comment. Therefore, we have made corrections in the abstract and conclusion. I hope they are sufficien. Page numer 1 line 19-31 and page numer 17 line 498-513
Comments 3: Introduction. Lines 73-80: Only hypotheses that were actually tested should be included. For example, the hypothesis regarding quantitative occurrence of melliferous vegetation was not tested. Melliferous vegetation is not synonym of number of plant species.
Response 3: Thank You for pointing this out. We agree with this comment. Melliferous vegetation is not synonym of number of plant species. Therefore, we have made corrections and we have included melliferous vegetation in the analyses. Page numer 4 Table 1, page numer 6 line 204, page numer 13 Figure 2 and page numer 14 Figure 3.
Comments 4: Materials and Methods Line 122: What do you mean by “phytosociological”?
Response 4: This term is commonly used in other English language publications. The fact is that in English-language articles on vegetation research using the Braun Blanquet method (1964) there are synonyms such as "relevés", "phytosociological record" and "vegetation relevés". But the phrase "phytosociological survey" is also often used, so we decide to keep the use of this term in our manuscript.
Comments 5: Lines 126-131: the explanation of how plant cover was determined is confusing. A better explanation should be given.
Response 5: We agree with the comments regarding the ambiguity of the methodological description in lines 126 - 131. The text has been corrected. Page numer 4 line 131-141.
Comments 6: Results Table1. Define in the Materials and Methods how you determined the stage of succession.
Response 6: Thank You for pointing this out. We have added information on how we classified transects into stages of succession. Page numer 5-6 line 190-196.
Comments 7: Table 5. Host plant Crataegus L? Where did authors obtain the host plant information from? I would suggest adding a column with references of studies that specifically refer to each of the butterfly species and host plants mentioned. For example, for Aporia crataegi L., Jugovic et al. (2017) https://doi.org/10.1007/s10841-017-9977-z, for Pieris rapae L., Badenes-Pérez (2023), https://doi.org/10.3390/plants12112148
Response 7: Thank You for pointing this out. We agree with this comment. Therefore, this literature item has been removed and replaced with literature with references to studies that specifically address each butterfly species. (Table 5), page numer 8-11,19-20 and line 276,569,572,585, 600-670
Comments 8: Some of the scientific names of butterfly species in Table 5 have missing parts. The full species name should be shown.
Response 8: Thank You for pointing this out. We have corrected it
Round 2
Reviewer 1 Report
The authors responded to my comments accordingly and made the necessary changes to the text of the MS. Nevertheless, I do not completely agree with their opinion concerning the Shannon-Wiener index, but here I acknowledge the authors' decision.
However, the Latin name of taxa should be presented properly. Therefore, each Latin name of the taxon (at generic and species level) must be, at least once, accompanied by the name of its author and the date of description.
Other comments were put directly into the MS text.

I have made several small corrections to the text.
Author Response
Response to Reviewer
Thank You very much for your positive review. We are glad that our corrections and clarifications were sufficient. I attach the manuscript with the corrections incorporated.
With best regards
Authors
